# Charge-Domain Static Random Access Memory-Based In-Memory Computing with Low-Cost Multiply-and-Accumulate Operation and Energy-Efficient 7-Bit Hybrid Analog-to-Digital Converter

Sanghyun Lee and Youngmin Kim *

School of Electronic and Electrical Engineering, Hongik University, Seoul 04066, Republic of Korea; mklo126@g.hongik.ac.kr
* Correspondence: youngmin@hongik.ac.kr

**Abstract:** This study presents a charge-domain SRAM-based in-memory computing (IMC) architecture. The multiply-and-accumulate (MAC) operation in the IMC structure is divided into current- and charge-domain methods. Current-domain IMC has high-power consumption and poor linearity. Charge-domain IMC has reduced variability compared with current-domain IMCs, achieving higher linearity and enabling energy-efficient operation with fewer dynamic current paths. The proposed IMC structure uses a 9T1C bitcell considering the trade-off between the bitcell area and the threshold voltage drop by an NMOS access transistor. We propose an energy-efficient summation mechanism for 4-bit weight rows to perform energy-efficient MAC operations. The generated MAC value is finally returned as a digital value through an analog-to-digital converter (ADC), whose performance is one of the critical components in the overall system. The proposed flash-successive approximation register (SAR) ADC is designed by combining the advantages of flash ADC and SAR ADC and outputs digital values at approximately half the cycle of SAR ADC. The proposed charge-domain IMC is designed and simulated in a 65 nm CMOS process. It achieves 102.4 GOPS throughput and 33.6 TOPS/W energy efficiency at array size of 1 Kb.

**Keywords:** in-memory computing; static random access memory; charge-domain SRAM–IMC; flash-SAR ADC; deep neural network

## 1. Introduction

Artificial intelligence (AI) is significantly contributing to technological progress as it can be applied to various fields. As a result, the amount of data to be processed is rapidly increasing [1]. However, in the von Neumann architecture, data movement overhead occurs because of limitations in separating memories and processors [2,3]. To address these limitations, in-memory computing (IMC) is considered the most promising next-generation computing architecture [4]. IMC has the advantage of significantly reducing the number of accesses to memory by directly performing computing work inside the memory. This can significantly reduce the energy consumption of well-known bottlenecks. The implementation of IMC is divided into digital and analog techniques. In the case of digital IMC, MAC operations are predominantly carried out using logic gates. However, this approach not only consumes a large area due to wide adder trees, but also requires more clock cycles since the operations are executed step by step. On the other hand, analog IMC achieves efficient MAC operations by leveraging Ohm's law and Kirchhoff's laws [5]. This results in reduced area consumption and allows for faster computation [6].

In an IMC environment, data stored in memory must be directly accessed and processed. Therefore, static random access memory (SRAM), which can read/write at high speed, is the best choice for implementing an IMC structure. In addition, unlike dynamic

random-access memory (DRAM), SRAM provides high endurance because no data is lost owing to the physical characteristics of the memory cell [7]. Therefore, SRAM is preferred in IMC environments where fast processing speed and energy efficiency are required. SRAM–IMC uses the current domain computing approach [8–16]. Generally, the current domain computing method using a 6T-SRAM bitcell enables high-speed calculation owing to parallel processing and fast calculation. However, the current domain computing approach has several limitations, as shown in Figure 1. First, a constant current must be discharged between bitcells sharing a bitline (BL). However, non-linearity may occur due to process variation. Second, the word lines (WL) of each bitcell sharing the BL can be activated simultaneously for logic operations, causing read disturbance problems. To address these problems, 8T-SRAM cells with separate read and write paths are used, as shown in Figure 2b [8]. However, 6T-SRAM cells feature two read paths (BL and BLB), while 8T-SRAM cells have only one read path (RBL), thereby limiting the available methods of computation. As illustrated in Figure 2c, ref. [12] utilizes 10T-SRAM cells to achieve two read paths without encountering read failures. Nevertheless, this approach results in a larger footprint due to the inclusion of additional transistors. Owing to the aforementioned challenges, current-domain computing continues to face issues of low accuracy in the inference stage, attributed to nonlinearity, and high power consumption resulting from dynamic computation.

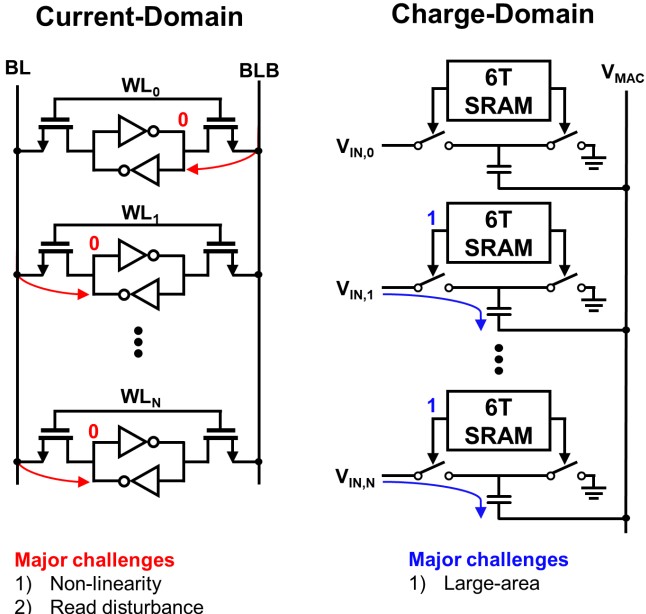

**Figure 1.** Comparison of current and charge-domain SRAM–IMC.

Recently, the SRAM–IMC of charge-domain computing has been reported to address the limitations of current-domain computing [17–27]. As shown in Figure 1, charge-domain computing uses capacitors added inside individual bitcells to perform analog multiplication. Although concerns about large areas caused by the capacitor inside the bitcell arise, this issue is addressed by placing metal–oxide–metal capacitors above the bitcell, eliminating the need for an extra area [18]. Additionally, analog multiplication of charge sharing method by capacitors inside each bitcell achieves higher linearity than that of the current domain [23]. However, the charge-sharing schemes reported in [18,19] are typically implemented using switched-capacitor structures and, as a result, require extra input signals. Additionally, the switching process may cause charge injection, potentially reducing computation accuracy. Figure 2d,e provide an overview of charge-domain bitcells. Although 10T1C bitcells [25] achieve high linearity in large arrays, they occupy a substantial individual cell area. The 8T1C bitcell utilized in the literature [27] occupies a relatively smaller area, but suffers from threshold voltage drop caused by NMOS access transistors.

In this study, we propose an SRAM-IMC structure employing capacitive coupling to address issues present in the existing switched capacitor-based systems. This configuration enables summation without the need for additional signals, and 4-bit calculations are executed through a capacitive voltage divider. Also, as shown in Figure 2f, the 9T1C bitcell is selected considering the trade-off between area and threshold voltage drop. The bitcell area is minimized by configuring the access transistor that transport the activation input as the transfer gate and the transistor that connects to the ground as a single NMOS.

### Current-domain

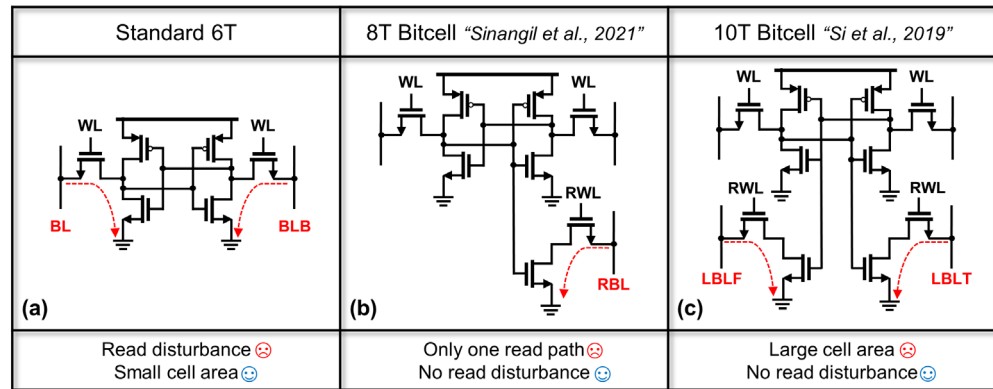

### Charge-domain

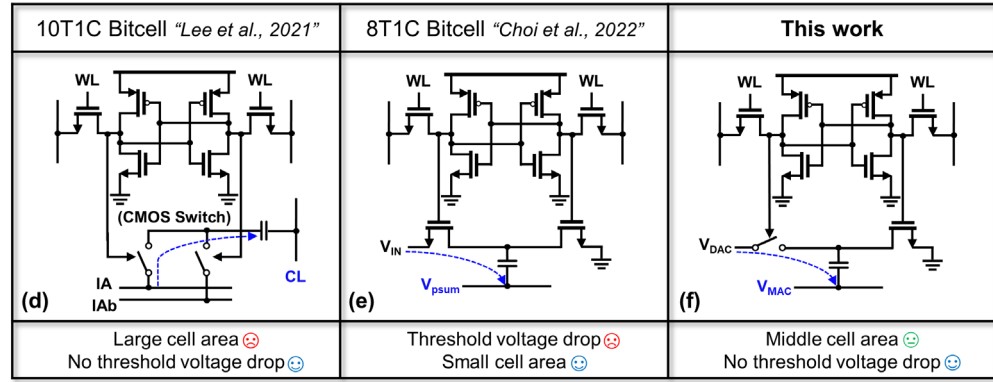

**Figure 2.** Structure of bitcell primarily used in current and charge and characteristics of each structure. Current-domain structures are shown in ((**a**,**b**) [8], and (**c**) [12]) and charge-domain structures are shown in ((**d**) [25], (**e**) [27], and (**f**)).

One of the key components in SRAM–IMC is the analog-to-digital converter (ADC), which converts the analog MAC results generated in the memory array into digital values. Among the various ADC types, the most widely used in SRAM–IMC are flash ADCs and successive approximation register (SAR) ADCs. Flash ADCs use multiple comparators to convert analog inputs to digital during one clock cycle. However, conventional flash ADCs require numerous comparators ($2^N - 1$, where N is the resolution) as resolution increases, resulting in significant area and power consumption. As a result, Flash ADCs need to allocate sufficient area and can only be used at low resolutions. Contrarily, SAR ADCs can achieve high resolution with a single comparator and relatively energy-efficient data conversion. Nonetheless, SAR ADCs are slow to convert because they process 1 bit per cycle. To address the distinct limitations of both ADC types, a hybrid structure termed flash-SAR ADC (FS ADC) has been proposed [28,29]. By combining the advantages of flash ADC and SAR ADC, FS ADC can output digital values in half the clock cycle compared with conventional SAR ADCs, alleviating the slow conversion speed of SAR ADCs. Additionally, energy efficiency can be maximized by utilizing a flash ADC for a fraction of the resolution.

Our main contributions are as follows:

- The proposed capacitive coupling is free from charge injection and additional input signals that occur based on switched-capacitor. Thus, the summation mechanism proposed in this paper can reduce the required capacitance substantially.
- Proposed an energy-efficient 7-bit flash-SAR Hybrid ADC (FS ADC) that merges the strengths of both flash and SRA ADC architectures to address the limitations associated with each ADC.
- The proposed coarse–fine architecture of the flash ADC reduces the number of comparators from $2^N - 1$ to $2^{N-1}$, minimizing the impact of the comparator's input gate capacitance. This not only reduces errors in the MAC generated by capacitive coupling in in-memory arrays, but also enables 4-bit summation with low capacitance.

The remainder of this paper proceeds as follows. Section 2 describes the proposed charge-domain computing architecture. Section 3 explains the proposed Flash-SAR Hybrid ADC. Simulation results are described in Section 4. Finally, Section 5 concludes the paper.

## 2. Proposed Charge-Domain Computing Architecture

In this study, the proposed SRAM–IMC employs a charge-domain methodology for digital–analog mixed-signal processing, performing the inner product operation between a binary weight vector and an input vector. This approach converts digital input signals into analog voltages and accumulates them continuously based on the values of the binary weight row, resulting in the computation of the inner product. This methodology offers low power consumption and high speed, while minimizing the number of logic operations required to calculate the inner product between the binary weight and input vector. Figure 3 shows the overall structure of the proposed charge-domain IMC. The structure consists of a $32 \times 32$ 9T1C array, an address decoder, thirty-two 4-bit input DACs, and eight flash-SAR ADCs. Each of the thirty-two 4-bit input DACs takes a 4-bit input and applies it to 32 binary weight columns, and the 32 binary weight rows compute the vector-matrix multiplication with the input DACs. Four weight rows perform the MAC operation using a 4-bit analog summation method. The flash-SAR ADCs convert analog 4-bit summation values received from the 9T1C array into 7-bit digital output values.

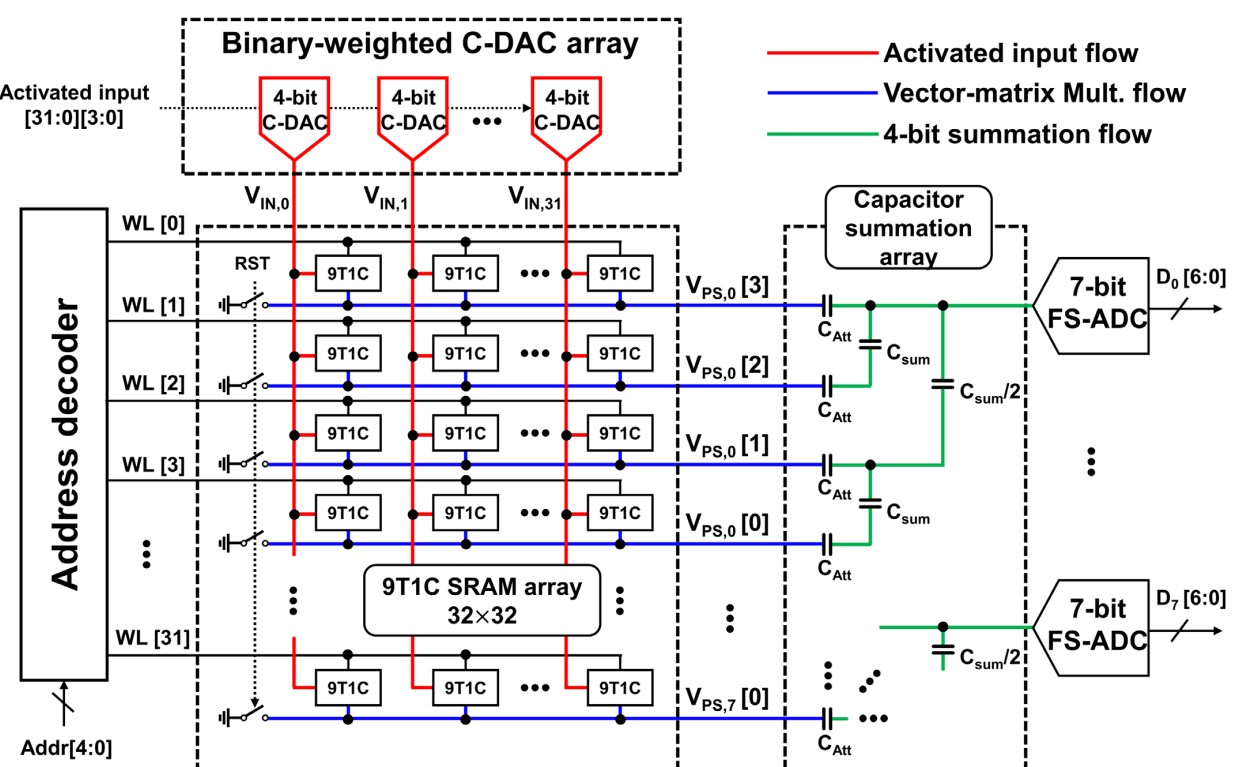

**Figure 3.** Structure of bitcell primarily used in current and charge and characteristics of each structure.

## 2.1. Binary Weighted Capacitor DAC

Figure 4 shows the 4-bit capacitor DAC used in the proposed structure. This DAC utilizes four capacitors to generate analog output values for 16 input codes ranging from $0000_{(2)}$ to $1111_{(2)}$. These capacitors are connected in descending order of size, with ratios of $8C_D$, $4C_D$, $2C_D$, and $C_D$, corresponding to the most significant bit (MSB) to the least significant bit (LSB) of a 4-bit input signal. The 4-bit capacitor DAC has both the top and bottom plates grounded in the initial stage and later generates $V_{IN}$ by the $DAC_{EN}$ signal. At $V_{DD}$ = 1 V, the voltage increment per 1 LSB is 62.5 mV, and among the 16 possible input codes, including 0, the 4-bit DAC can represent the highest voltage of 937.5 mV. The generated $V_{IN}$ is provided as input to the 9T1C bitcell.

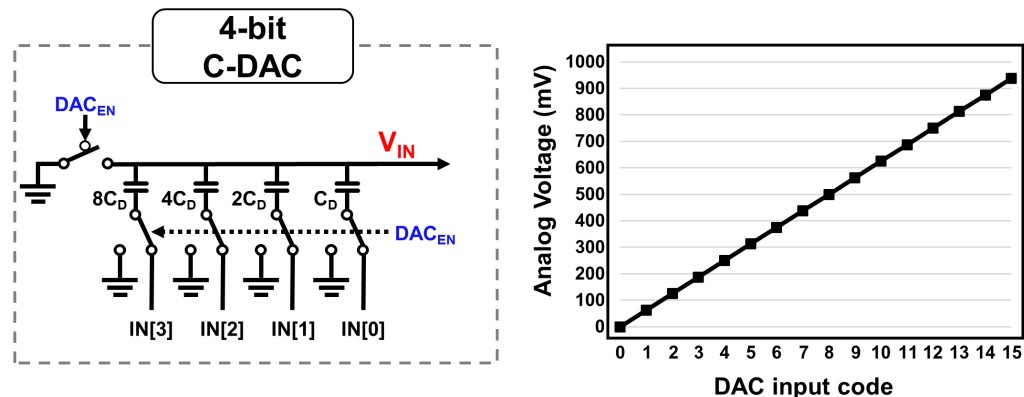

**Figure 4.** Structure of capacitor-based DAC (**left**) and the ideal linearity plot (**right**).

## 2.2. 9T1C Bitcell

Figure 5 shows the 9T1C bitcell used in the proposed architecture. It comprises a 6T-SRAM cell for weight storage, a transmission gate, and an NMOS controlled by the stored weight. Additionally, it includes a capacitor ($C_{cell}$ = 1.3 $f$F) that performs multiplication operations within a single bitcell. When the Q value of the bitcell is 0, Q_B stores the opposite value 1, and the NMOS is activated by Q_B, causing the top plate of $C_{cell}$ to be grounded. Conversely, when the Q value of the single bitcell is 1, the Q value is applied to the gate of the NMOS that forms the transmission gate, whereas 0 is applied to the gate of the PMOS. This activation of the transmission gate enables $V_{IN}$ to be transferred to the top plate of $C_{cell}$ without any voltage drop, sending it to the $V_{PS}$ row. The individual inputs and weights of the bitcells perform multiplication within $C_{cell}$ coupled with the $V_{PS}$ row. Moreover, all $C_{cell}$ values coupled with the row are accumulated in the $V_{PS}$ row through charge redistribution, as expressed in

$$V_{PS} = \frac{\sum_{i=0}^{31} V_{IN,i} \times C_{cell,i}}{32 \times C_{cell}}. \tag{1}$$

## 2.3. Proposed Capacitor Summation Mechanism

In [27], the total capacitance of the bridge capacitors used for 4-bit summation is remarkably high, reaching 272 times that of a single-bit cell capacitor. Given that a capacitor's charge/discharge current is directly proportional to its size, an increase in size leads to higher power consumption. Consequently, reducing the size of the capacitor used for summation becomes crucial for mitigating power consumption. The proposed structure introduces a capacitive coupling summation method, depicted in Figure 6, which diverges from the approach presented in [27]. The summation mechanism involves VPS [3] and VPS [2], each contributing in an 8:4 ratio in the capacitive voltage divider due to capacitive coupling by $C_{sum}$ = 32 × $C_{cell}$ ‖ $C_{Att}$, resulting in a cumulative ratio of 2:1. Similarly, VPS [1] and VPS [0] are combined in a 2:1 ratio, and the sum of VPS [3–2] and VPS [1–0] is aggregated in a 4:1 ratio using the capacitor in $C_{sum}$/2. Finally, $V_{MAC}$, representing the 4-bit sum value in the ratio 8:4:2:1, is transmitted to the FS ADC. The significance of

$C_{Att}$ (23.1 $f$F) lies in its effective reduction in $C_{sum}$'s capacitance by connecting directly with 32 $C_{cell}$ in parallel to $V_{PS}$. This results in the capacitance of $C_{sum}$ being equal to the series combined capacitance of 32 $C_{cell}$ and $C_{Att}$. The total capacitance required for 4-bit summation is approximately 53 times that of a single bit cell capacitor, a significantly lower value compared to previous studies. Capacitance of $C_{Att}$ is carefully set to a level that ensures the total capacitance required for summation does not lead to errors caused by the gate capacitance of the FS ADC. While directly connecting summing capacitors to the ADC input might cause errors due to low total capacitance, this is acceptable in our proposed FS ADC due to its low gate capacitance, as discussed in Section 3.1.

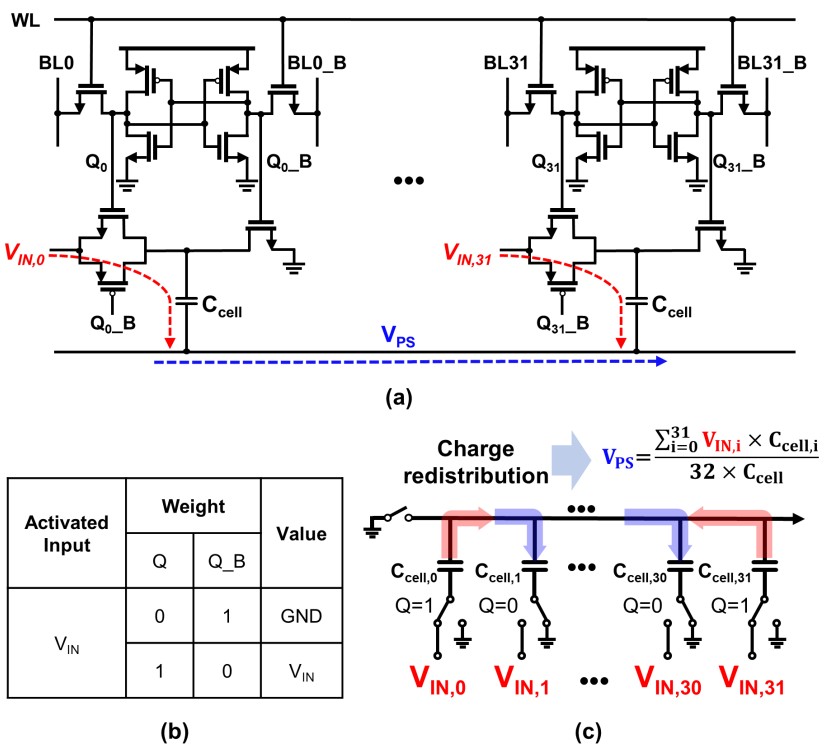

**Figure 5.** (**a**) 9T1C bitcells multiplication, (**b**) truth table, and (**c**) analog accumulation process by charge redistribution.

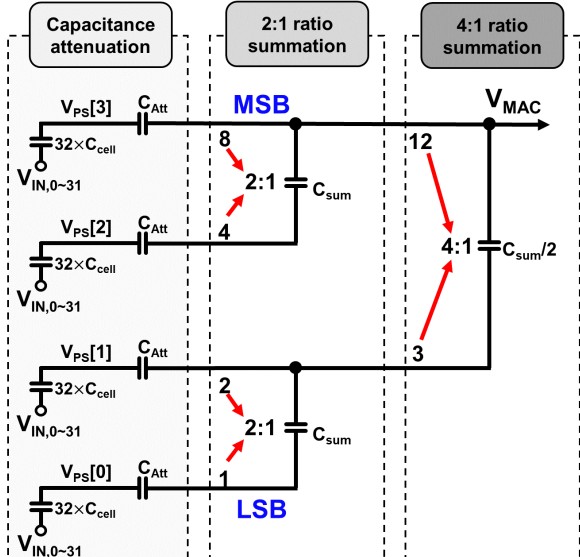

**Figure 6.** 8:4:2:1 capacitor summation mechanism of 4-bit weighted rows.

## 3. Architecture of Flash-SAR Hybrid ADC

Figure 7 shows the structure of the proposed 7-bit FS ADC, comprising a 3-bit coarse-fine flash ADC, SAR logic, a 7-bit C-DAC, thermometer to binary encoder (TM2B), and an output register. Initially, the $V_{MAC}$ formed by the summation capacitors is input to the coarse-fine flash ADC for conversion of the upper 3-bit. Once the conversion is completed, the MSB of the upper 3-bit, which is also the MSB stage of the entire ADC, is stored in the output register while being connected to the $4C_U$ of the C-DAC in the FS ADC. The remaining 2-bit of the coarse-fine flash ADC are connected to the capacitor units ($C_U$s) of the MSB stage in a thermometer code format before passing through the TM2B. This method prevents any delay in the start of the SAR ADC for the lower 4-bit. This delay would otherwise have occurred if the TM2B were connected to the C-DAC after going through SAR logic. A detailed description of the coarse–fine flash ADC is discussed in Section 3.1.

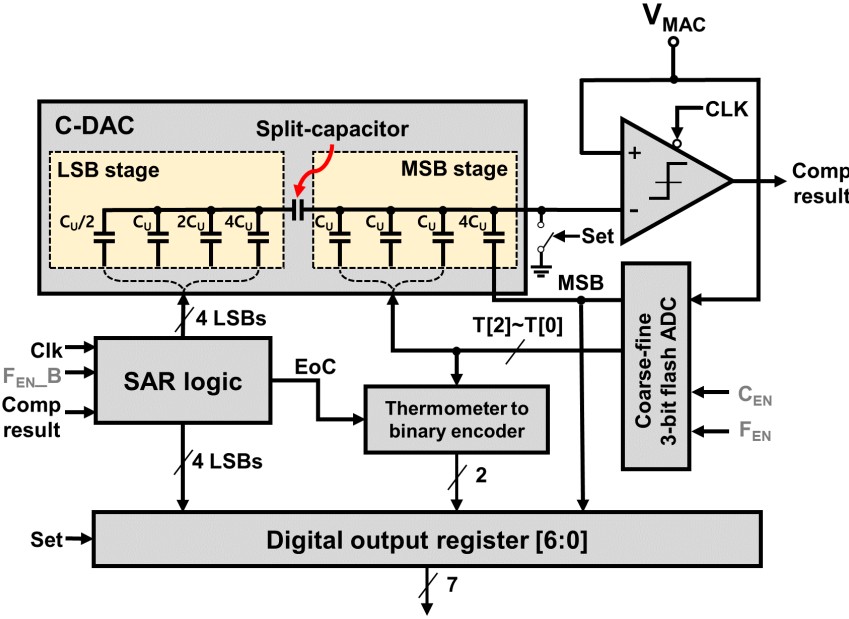

**Figure 7.** Overall structure of proposed FS ADC.

To address the challenge of implementing a 7-bit C-DAC, which requires large capacitors of approximately $128C_U$ as the resolution increases, the proposed FS ADC employs divided capacitors to limit the capacitor size used in the C-DAC to a maximum of $4C_U$. Moreover, as the FS ADC must differentiate between the MSB and LSB stages, the use of divided capacitors becomes an appropriate choice from various aspects.

Figure 8 shows the timing diagram of the FS ADC, including the operation of the IMC. Initially, the RST signal grounds both the top and bottom plates of the summation capacitor used in MAC operation to reset $V_{MAC}$. Simultaneously, during RST = 1, the $DAC_{EN}$ signal is temporarily held at 0. When RST = 0, 4-bit activation inputs for MAC operation are supplied to each column to generate $V_{MAC}$. Subsequently, the SET signal becomes 0, preparing to form a reference voltage in the C-DAC, and at the same time, the coarse–fine flash ADC operation begins by $C_{EN}$. Once the comparison results for the coarse stage are obtained, forming the reference voltage for the fine stage, the $F_{EN}$ rises to 1, starting the fine stage comparison (coarse–fine flash ADCs are discussed in detail in Section 3.1). Simultaneously, through the $F_{EN\_B}$, the SAR logic forms the reference voltage on the C-DAC. Subsequently, when the conversion for the lower 4-bit is completed via the SAR comparator operating on the falling edge, the End-of-Converting (EoC) signal from the SAR logic triggers TM2B, ultimately storing a 7-bit digital value in the output register. Additionally, to ensure a stable settling time of the MSB stage $C_U$, the frequency of the FS ADC is limited to 500 MHz. The overall IMC structure can reset and generate the $V_{MAC}$ result within one clock cycle of the FS ADC. Therefore, the 50 MHz frequency represents

the operational frequency of the entire IMC structure, including resetting and storing the ADC output in registers. If the FS ADC operates at a lower frequency, it may provide more stability, but considering the potential decrease in the throughput of the overall architecture, the frequency of the FS ADC is determined with this aspect in mind.

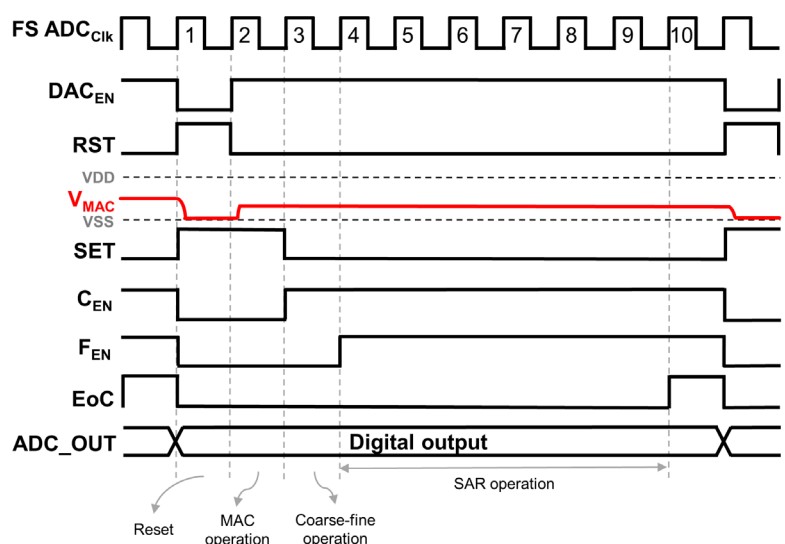

**Figure 8.** Timing diagram of FS ADC with IMC array.

### 3.1. Coarse–Fine Flash ADC

Conventional flash ADCs that require $2^N - 1$ comparators also use a significant number of comparators at low bits. Furthermore, owing to the summation capacitor attenuation mentioned in Section 2.3, $V_{MAC}$, the input of the FS ADC, can be reduced by the parasitic capacitance of the metal and the input gate capacitance of the comparator. Thus, the choice of resolution for the flash ADC, which requires many comparators, becomes a critical issue in the design of the FS ADC. In the proposed FS ADC, the upper 3-bit is controlled by the flash ADC, and to minimize the number of comparators, a coarse–fine structure is adopted to perform flash ADC operation with only $2^{N-1}$ comparators. As only four comparators are consumed, the effect of the gate capacitance on the $V_{MAC}$ formed by the summation capacitor can be minimized.

Figure 9 shows the structure of the 3-bit coarse–fine flash ADC. The voltage divider, designed as a R-ladder structure, forms a total of seven reference voltages from [6] to [0]. In the coarse stage, the VMAC formed by the summation capacitors is fed as an input to the coarse comparators, and its comparison with [3] determines the MSB of the 3-bit flash ADC. Then, based on the comparison result of the coarse comparators, the reference voltages for the three fine comparators are determined and the conversion for the remaining 2-bit begins. When VMAC < [3], the output of the coarse comparators is 0 and the reference voltages for the three fine comparators are set to [2]–[0]. Conversely, when VMAC > [3], the output of the coarse comparators is 1 and the reference voltages for the three fine comparators are set to [7]–[5]. As the MSB of the flash ADC is identical to the MSB of the entire FS ADC structure, it is directly stored in the output register while the remaining 2-bit is passed on to the next stage in the form of a thermometer code.

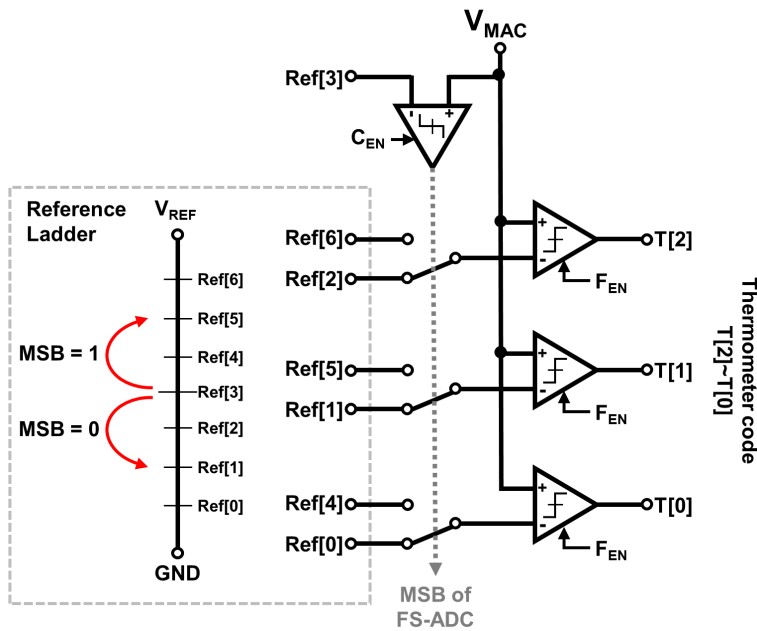

**Figure 9.** Block diagram of a 3-bit coarse–fine flash ADC.

### 3.2. High-Speed Rail-to-Rail Comparator

Conventional comparators typically use a differential input pair consisting of NMOS transistors for comparison. However, in some cases, when the reference voltage closest to GND does not exceed the $V_{TH}$ of the input gate, it may result in incorrect output values. This is also the case when $V_{MAC}$ fails to exceed $V_{TH}$. Therefore, to support data conversion across the entire dynamic range, the proposed FS ADC requires comparators that can perform rail-to-rail detection and high-speed operation. Figure 10a shows the comparator proposed in the literature [30]. It utilizes both NMOS and PMOS differential input pairs, enabling operation over the full dynamic range and achieving sufficient speed, rendering it suitable for the proposed architecture. However, because the number of input pairs is higher than when using only NMOS, kickback noise due to gate capacitance increases. To minimize kickback noise, the sizes of the differential input pairs consisting of NMOS and PMOS are designed using the minimum size in the 65 nm CMOS process. In addition, to prevent comparison result loss caused by CLK, a latch is added to ensure accuracy.

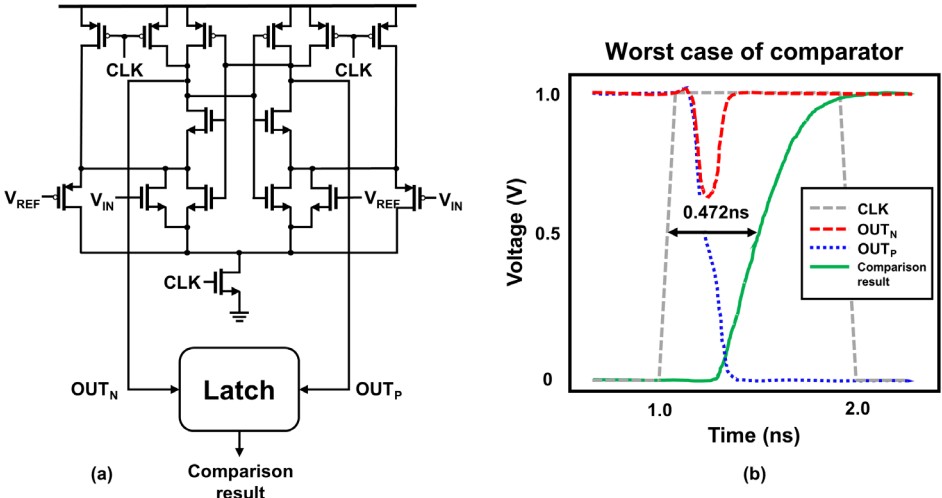

**Figure 10.** (**a**) Rail-to-rail sensing comparator [30] added latch stage, (**b**) the simulated worst-case delay.

Figure 10b shows the simulation result of the comparator with the latch under worst-case. The simulation of the worst-case delay is performed from the time when CLK becomes

$V_{DD}/2$ until the comparison result value becomes $V_{DD}/2$. The simulation result shows that the worst-case delay is approximately 0.472 ns, and the result can be obtained within half of the 500 MHz CLK period used in the FS ADC.

## 4. Simulation Results

In this section, we describe the simulation results conducted at the pre-layout stage to evaluate the performance of the proposed charge-domain IMC structure. The proposed structure is designed using a 65 nm CMOS process and operated at $V_{DD}$ = 1 V. To achieve a capacity of 1 Kb, a 32 × 32 array of 9T1C bitcells is constructed and the operating frequency of the IMC is set to 50 MHz. Additionally, the internal SAR ADC within the FS ADC is operated at a frequency of 500 MHz, ultimately converting the analog signal into a 7-bit digital output value.

The proposed SRAM-IMC mainly operates with capacitors, and the biggest impact is capacitor mismatch. However, capacitors with metal-oxide-metal (MoM) structure possess very good matching properties [31,32]. As reported in [18], the capacitor mismatch for in-memory computing using MoM capacitors in 65 nm CMOS was found to be only 1%. This is lower than the quantization error of a 7-bit ADC. Additionally, while the charge sharing method must consider the charge injection and capacitor mismatch characteristics due to the MOS switch, the proposed method can only consider the capacitor mismatch. Therefore, high linearity can be expected even after post-layout and chip measurement.

### 4.1. Linearity of 4-bit DAC

The proposed charge-domain SRAM–IMC structure forms a path between the individual bitcells, the summation and 4-bit input DAC, which can cause output voltage variations using coupling capacitors. Moreover, with additional paths between the weight row and each bitcell, a 4-bit input DAC may induce poor linearity. Figure 11a shows the linear graph of the 4-bit input DAC when the FS ADC outputs the final digital values in the overall IMC structure. Moreover, we conducted experiments with all weight values in the weight row set to 1 to explore the worst-case. The worst-case occurred when the 4-bit input value was $1111_{(2)}$, resulting in approximately 5% distortion compared with the ideal analog value of the 4-bit DAC.

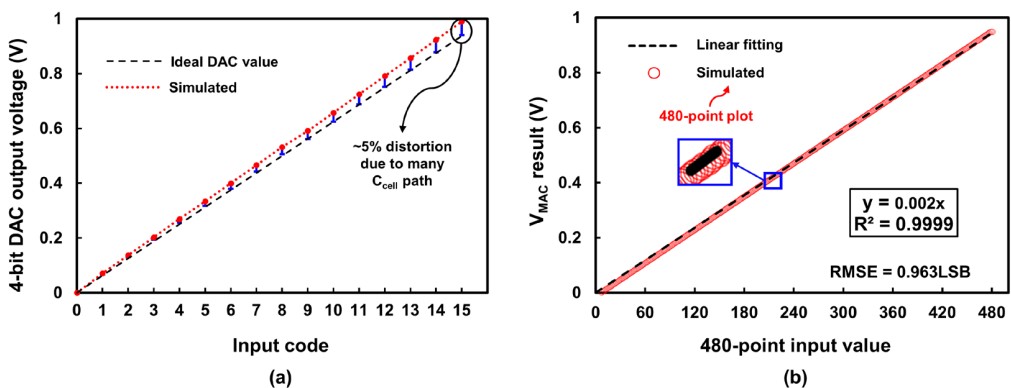

**Figure 11.** (**a**) Worst-case distortion simulation of 4-bit input DAC, (**b**) linearity of $V_{MAC}$ for 480-point linear inputs from 4-bit DAC.

### 4.2. Linearity of Charge-Domain MAC Operation

In the proposed IMC structure, MAC operations are finalized through a 4-bit summation for each row group. Figure 11b shows the transfer curve of $V_{MAC}$, computed using thirty-two 4-bit input DACs and summation capacitors. Additionally, the weight of the entire array is stored as 1 to simulate all possible values that $V_{MAC}$ can represent. The simulation consisted of sequentially incrementing the first 4-bit input DAC from $0000_{(2)}$ to $1111_{(2)}$, and then following the same increasing pattern for the subsequent 4-bit input DACs. This process generated a total of 480 $V_{MAC}$ outputs corresponding to the 32 input DACs.

The charge-domain MAC operation in the proposed design exhibits excellent linearity with a coefficient of determination ($R^2$) of 0.9999. The root mean square error (RMSE) represents the average error between the expected $V_{MAC}$ results using Equation (1) and the actual simulated values. The simulated RMSE shows a significantly satisfactory value of 0.963 LSB, which remains less than 1.

### 4.3. Performance of FS ADC

The R-ladder of the FS ADC utilizes resistive voltage division, and the selection of resistor values is closely related to the overall performance. Larger resistor values lead to lower power consumption, but linearities decrease because of kickback noise. Conversely, smaller resistor values result in higher power consumption but increased linearity. Figure 12a shows the dynamic power consumption with respect to resistor values. In the 7-bit FS ADC, resistor values were selected to prevent missing digital output values. Considering the trade-off between power consumption and linearity, the proposed FS ADC adopts a resistance of 500 $\Omega$.

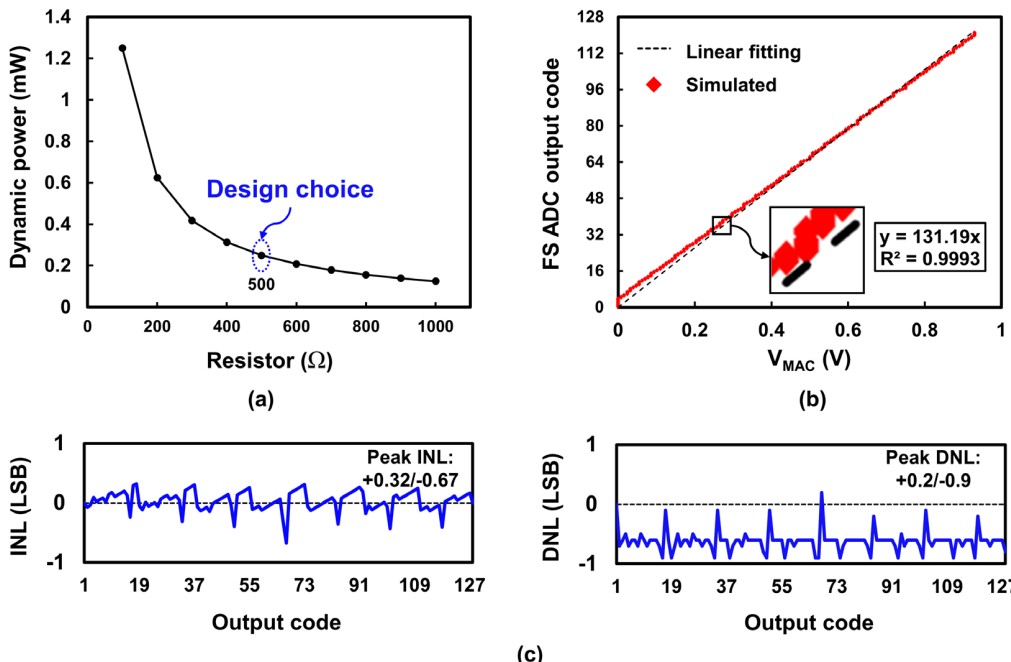

**Figure 12.** (**a**) Dynamic power dissipation versus resistor value, (**b**) 480-point $V_{MAC}$ analog value versus output code of FS ADC, and (**c**) INL and DNL of proposed FS ADC.

Figure 12b illustrates the final output of the entire process, from the 4-bit C-DAC to the FS ADC, by combining the FS ADC with the memory array. In other words, Figures 11a and 12b represent results obtained through the same simulation over a long period of time. The resulting R value is 0.9993, demonstrating a linearity similar to the $R^2$ value of the previous $V_{MAC}$.

The performance evaluation of the FS ADC was conducted using both the linearity assessment of the MAC operation and the final digital output values obtained through the ADC. Figure 12c shows the simulated results of integral non-linearity (INL) and differential non-linearity (DNL) for the FS ADC. Ideal DACs were connected and extracted using Cadence Spectre to convert the final digital output values into analog values. The extracted analog output values were computed in MATLAB R2017b, and the evaluated max/min values for INL and DNL were +0.32/−0.67 and +0.2/−0.9, respectively.

### 4.4. Performance Comparison

The proposed charge-domain SRAM–IMC operates in parallel with 32 inputs and 32 individual weight rows at a maximum frequency of 50 MHz. The throughput (GOPS)

is calculated as $2 \times 32 \times 32 \times 1/(20 \text{ ns}) = 102.4$. The factor of 2 is multiplied to account for processing one MAC operation, which involves one multiplication and one addition performed simultaneously. For energy efficiency, the weights of all individual bitcells were set to 1, and the power consumptions for all possible cases represented by the thirty-two 4-bit input DACs were simulated. The average power consumption was 3.04 mW, achieving an energy efficiency of 33.6 TOPS/W. In the proposed architecture, the dominant power consumption is due to the R-ladder. The dynamic power consumption of eight R-ladders, each corresponding to an ADC, is 2 mW, accounting for approximately 66% of the total power. For example, if the array size of the proposed architecture is expanded to $128 \times 128$, 32 ADCs are used, and 32 R-ladders consume 8 mW of dynamic power. When R-ladder is accounted for, the average power consumption is calculated to be 12.12 mW, and the throughput and TOPS/W are 1638.4 GOPS and 135.2 TOPS/W, respectively.

Table 1 compares the performance of the proposed charge-domain IMC with those of previous studies. To ensure an accurate comparison, we utilized Figure of Merit (FoM) based on input precision $\times$ weight precision $\times$ energy efficiency (scaled to 65 nm) [33]. Despite the relatively smaller $32 \times 32$ array size of the proposed structure than those of previous studies, it demonstrates superior performance in terms of FoM.

**Table 1.** Performance comparisons with previous works.

| Parameter | This Work * | JSSC'19 [12] | JSSC'20 [16] | TCAS'23 [17] | JSSC'20 [21] | JSSC'21 [23] |
|---|---|---|---|---|---|---|
| Technology | **65-nm** | 65-nm | 65-nm | 28-nm | 65-nm | 65-nm |
| Bitcell Structure | **9T1C** | 10T | 12T | 6T | 10T1C | 6T |
| Array Size | **$32 \times 32$** | $256 \times 64$ | $256 \times 64$ | $128 \times 128$ | $2304 \times 256$ | $512 \times 256$ |
| Supply Voltage | **1 V** | 0.8~1.2 V | 0.6~1.0 V | 0.6~0.9 V | 0.85/1 V | 1.2 V |
| Frequency | **50 MHz** | 5 MHz | 100 MHz | 50 MHz | 100 MHz | N/A |
| Computing Type | **Charge** | Current | Current | Charge | Charge | Charge |
| ADC Type | **Flash-SAR** | Integrating | Flash | Flash | SAR | ciSAR ADC |
| Bit Precision (Input/Weight/Output) | **4/4/7** | 6/1/7 | 1/1/3.46 | 4/4/4 | 1/1/8 | 4/1/7 |
| Throughput ** (GOPS) | **102.4** | 8 | N/A | 204.8 | 2185 | 573.4 |
| Energy Efficiency(TOPS/W) | **33.6** | 40.3 | 403 | 16.9 (3.13) *** | 192 | 49.4 |
| FoM **** | **537.6** | 241.8 | 403 | 50 | 192 | 197.6 |

* Pre-layout simulation result. ** One MAC is counted as two operations (1 multiplication + 1 accumulation). *** Scaled to 65-nm, assume energy $\propto$ (Tech.)$^2$ [12]. **** FoM = input precision $\times$ weight precision $\times$ energy efficiency (scaled to 65-nm) [33].

## 5. Conclusions

The proposed SRAM–IMC performed MAC operations based on charge and achieved a reduced total capacitance for summation by utilizing capacitors. Additionally, to address the limitations of flash and SAR ADCs, we proposed an energy-efficient FS ADC that combines both ADCs, resulting in approximately half the cycle time compared with conventional SAR ADCs. Although a smaller total capacitance for summation could render the IMC sufficiently sensitive to noise when utilized as an input for the ADC, we minimize noise by employing a coarse-fine flash ADC inside FS ADC. The proposed structure is designed using a 65 nm CMOS process and simulated at a 1 V operating voltage. For the linear input of the 4-bit DAC, the proposed summation capacitor mechanism achieved high linearity and the data conversion in the FS ADC also demonstrated high accuracy. Despite having a smaller array size than those previous studies, the overall proposed structure achieves a throughput of 102.4 GOPS and an energy efficiency of 33.6 TOPS/W. Continuous

active research is ongoing to further explore the applications of IMCs, and we anticipate that our proposed structure will be a promising alternative solution.

**Author Contributions:** Conceptualization, S.L. and Y.K.; methodology, S.L.; software, S.L.; validation, S.L. and Y.K.; investigation, S.L.; resources, Y.K.; writing—original draft preparation, S.L.; writing review and editing, S.L. and Y.K.; visualization, S.L.; supervision, Y.K.; project administration, Y.K.; funding acquisition, Y.K. All authors have read and agreed to the published version of the manuscript.

**Funding:** This research was supported by the Ministry of Science and ICT (MSIT), Republic of Korea.

**Data Availability Statement:** All data underlying the results are available as part of the article and no additional source data are required.

**Acknowledgments:** This research was supported by the MSIT (Ministry of Science and ICT), Korea, under the ITRC (Information Technology Research Center) support program (IITP-2023-RS-2022-00156225) supervised by the IITP (Institute for Information & Communications Technology Planning & Evaluation). The EDA tool was supported by the IC Design Education Center (IDEC), Korea.

**Conflicts of Interest:** The authors declare no conflicts of interest.

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
