# Peer review of "Charge-Domain Static Random Access Memory-Based In-Memory Computing with Low-Cost Multiply-and-Accumulate Operation and Energy-Efficient 7-Bit Hybrid Analog-to-Digital Converter"

_electronics, doi:10.3390/electronics13030666_

Round 1

Reviewer 1 Report

Comments and Suggestions for Authors

It is clear that in-memory computing is a promising method in Artificial intelligence, and can address the bottleneck caused by von Neumann architecture by reducing the movement of data and providing high energy-efficient operations. In this manuscript, 9T1C bit cell SRAM architecture and FS ADC are proposed for charge-domain SRAM-based in-memory computing. The manuscript is well organized and this work is of some practical significance. However, some details should be further explored. 

  1. Explain the reason why the operating frequency of the IMC is finally set to 50 MHz with the fact that the FS ADC proposed in this manuscript can be operated under 500 MHz. Whether a low frequency like 50 MHz or 100 MHz ADC is enough to be suitable for this 9T1C bit cell array?
  2. The function of the signals in Figure 8 is unclear, better to add more description.
  3. In section 4.2, what’s the condition of cell weights, and number of SRAM cells operating for the MAC simulation?
  4. Is there any simulation result combining FS ADC with the memory array?
  5. Details need to be checked, like if Equation (1) is with VDD, etc.
Comments on the Quality of English Language

None

Author Response

Thank you very much for all the helpful comments from reviewers. We have revised the manuscript according to the reviewers’ comments and the changes are highlighted in red in the revision.

Comments and Suggestions for Authors

It is clear that in-memory computing is a promising method in Artificial intelligence, and can address the bottleneck caused by von Neumann architecture by reducing the movement of data and providing high energy-efficient operations. In this manuscript, 9T1C bit cell SRAM architecture and FS ADC are proposed for charge-domain SRAM-based in-memory computing. The manuscript is well organized and this work is of some practical significance. However, some details should be further explored.

  • Explain the reason why the operating frequency of the IMC is finally set to 50 MHz with the fact that the FS ADC proposed in this manuscript can be operated under 500 MHz. Whether a low frequency like 50 MHz or 100 MHz ADC is enough to be suitable for this 9T1C bit cell array?

Response: Thanks for your comments. In the proposed FS ADC, the comparator's response time is capable of operating even faster than 500 MHz, with a worst-case time of approximately 0.472 ns. However, to ensure the settling time of the MSB stage's CU reliably, the operation speed is restricted to 500 MHz. Additionally, the FS ADC consumes 9 clock cycles after receiving the VMAC input. The overall IMC structure can reset and generate the VMAC result within one clock cycle of the FS ADC. Therefore, the 50 MHz frequency represents the operational frequency of the entire IMC structure, including resetting and storing the ADC output in registers.

These details have been added to Section 3 as follows:

  • If the FS ADC operates at a lower frequency, it may provide more stability, but considering the potential decrease in the throughput of the entire IMC structure, the frequency of the FS ADC is determined with this aspect in mind.

  • The function of the signals in Figure 8 is unclear, better to add more description.

Response: Thank you for your comment. Per your comments, we have added explanations of the signals to clarify Figure 8.

These details have been added to the section 3 as follows:

  • Figure 8 shows the timing diagram of the FS ADC, including the operation of the IMC. Initially, the RST signal grounds both the top and bottom plates of the summation capacitor used in MAC operation to reset VMAC. Simultaneously, during RST = 1, the DACEN signal is temporarily held at 0. When RST = 0, 4-bit activation inputs for MAC operation are supplied to each column to generate VMAC. Subsequently, the SET signal becomes 0, preparing to form a reference voltage in the C-DAC, and at the same time, the coarse-fine flash ADC operation begins by CEN. Once the comparison results for the coarse stage are obtained, forming the reference voltage for the fine stage, the FEN rises to 1, starting the fine stage comparison (coarse-fine flash ADCs are discussed in detail in Section 3.1). Simultaneously, through the FEN_B, the SAR logic forms the reference voltage on the C-DAC. Subsequently, when the conversion for the lower 4-bit is completed via the SAR comparator operating on the falling edge, the End-of-Converting (EoC) signal from the SAR logic triggers TM2B, ultimately storing a 7-bit digital value in the output register. Additionally, to ensure a stable settling time of the MSB stage CU, the frequency of the FS ADC is limited to 500 MHz. The overall IMC structure can reset and generate the VMAC result within one clock cycle of the FS ADC. Therefore, the 50 MHz frequency represents the operational frequency of the entire IMC structure, including resetting and storing the ADC output in registers.

  • In section 4.2, what’s the condition of cell weights, and number of SRAM cells operating for the MAC simulation?

Response: Thanks for your comments and sorry for the confusion. To simulate all possible values that VMAC can represent, the weight values of all 9T1C bitcells are stored as 1. For instance, if out of the 32 bitcells connected in one VPS row, 16 bitcells store a weight value of 0, the range of values that the VPS row can represent would be halved. For this reason, the simulation was conducted with all weights stored as 1.

These details have been added to Section 4.2 as follows:

  • Additionally, the weight of the entire array is stored as 1 to simulate all possible values that VMAC can represent.

  • Is there any simulation result combining FS ADC with the memory array?

Response: Thanks for your comments and sorry for the confusion. Figure 12b illustrates the final output of the entire process, from the 4-bit C-DAC to the FS ADC, by combining the FS ADC with the memory array. In other words, Figures 11a and 12b represent results obtained through the same simulation over a long period of time. Similarly, Figure 11b, as a result connected to the FS ADC, includes gate capacitors and kickback noise present at the input stage of the FS ADC.

The above content has been summarized and added to Section 4.3 as follows.

  • Figure 12b illustrates the final output of the entire process, from the 4-bit C-DAC to the FS ADC, by combining the FS ADC with the memory array. In other words, Figures 11a and 12b represent results obtained through the same simulation over a long period of time.

  • Details need to be checked, like if Equation (1) is with VDD, etc.

Response: Thanks for your comments and sorry for the confusion. Equation (1) has been confirmed to repeatedly use VDD. In this equation, VIN represents the analog voltage value generated from the C-DAC, and it is already multiplied by the VDD value.

Equation (1) was modified as follows.

Reviewer 2 Report

Comments and Suggestions for Authors

The paper presents detailed implementation and schematic-level simulation results corresponding to an 9T1C SRAM in-memory-compute architecture. Generally speaking, the paper is very well written and the figures corresponding to both the simulation results, as well as the overall architecture explaning the scheme are clear and adequate.

Minor typos (SRA-ADC instead of SAR ADC) can be corrected using a through reading.

It would be nice if the authors could include a post-layout and tapeout projection of the performance numbers. This would make the paper very relevant to compasion with other architectures / schemes proposed in literature and compared with in the comparison  table. 

Other than that, I have no further significant comments. 

Author Response

Thank you very much for all the helpful comments from reviewers. We have revised the manuscript according to the reviewers’ comments and the changes are highlighted in red in the revision.

Comments and Suggestions for Authors

The paper presents detailed implementation and schematic-level simulation results corresponding to an 9T1C SRAM in-memory-compute architecture. Generally speaking, the paper is very well written and the figures corresponding to both the simulation results, as well as the overall architecture explaning the scheme are clear and adequate.

  • Minor typos (SRA-ADC instead of SAR ADC) can be corrected using a through reading.

Response: Thanks for your comments. We have corrected the typos throughout the manuscript.

  • It would be nice if the authors could include a post-layout and tapeout projection of the performance numbers. This would make the paper very relevant to compasion with other architectures / schemes proposed in literature and compared with in the comparison

Response: Thanks for your comments. The proposed in-memory computing is mainly affected by operations related to capacitors, and the main impact is the mismatch of the capacitors. The proposed IMC was implemented using Metal-Oxide-Metal (MoM) in a 65nm CMOS process. The capacitor mismatch of MoM capacitors in 65-nm was found to be only 1% [18], which is lower than the quantization error of a 7-bit ADC. Additionally, in the charge-sharing scheme, charge injection occurs due to the MOS switch, whereas in the proposed scheme, only the capacitor mismatch is considered, thus, much better linearity can be expected. In addition, we hope to present new results, since post-layout simulation and tape-out is being conducted.

The above content has been summarized and added to Section 4 as follows.

  • The proposed SRAM-IMC mainly operates with capacitors, and the biggest impact is capacitor mismatch. However, capacitors with metal-oxide-metal (MoM) structure possess very good matching properties [32], [33]. As reported in [18], the capacitor mismatch for in-memory computing using MoM capacitors in 65-nm CMOS was found to be only 1%. This is lower than the quantization error of a 7-bit ADC. Additionally, while the charge sharing method must consider the charge injection and capacitor mismatch characteristics due to the MOS switch, the proposed method can only consider the capacitor mismatch. Therefore, high linearity can be expected even after post-layout and chip measurement.

Reviewer 3 Report

Comments and Suggestions for Authors

This paper presents a SRAM-based IMC architecture with reduced variability and non-linearity, which ensures  102.4 GOPS throughput and 33.6 TOPS/W energy efficiency with 1 kb array.

The paper is well written. The introduction is complete and results are attractive. I have just few suggestions for the Authors:

1) Could the Authors expand the Introduction with a brief comparison between Analog In-Memory Computing and Digital In-memory Computing? Some key-references are listed hereafter.

[1] P. Yao, DOI: 10.1038/s41586-020-1942-4;

[2] A. Antolini, DOI: 10.1109/ESSCIRC55480.2022.9911447;

[3] I. Muñoz-Martín, doi: 10.1109/TED.2021.3118996.

2) Could the Authors provide an estimation of throughput and GOPS/W with larger size of the array (i.e., with a mathematical model and/or numerical simulations?

Author Response

Thank you very much for all the helpful comments from reviewers. We have revised the manuscript according to the reviewers’ comments and the changes are highlighted in red in the revision.

Comments and Suggestions for Authors

This paper presents a SRAM-based IMC architecture with reduced variability and non-linearity, which ensures 102.4 GOPS throughput and 33.6 TOPS/W energy efficiency with 1 kb array.

The paper is well written. The introduction is complete and results are attractive. I have just few suggestions for the Authors:

  • Could the Authors expand the Introduction with a brief comparison between Analog In-Memory Computing and Digital In-memory Computing? Some key-references are listed hereafter.

[1] P. Yao, DOI: 10.1038/s41586-020-1942-4;

[2] A. Antolini, DOI: 10.1109/ESSCIRC55480.2022.9911447;

[3] I. Muñoz-Martín, doi: 10.1109/TED.2021.3118996.

Response: Thanks for your comments. Per your comment, we have complemented the details in Introduction for better understanding. Even though the papers listed are not SRAM-based in-memory computing, we have reviewed and analyzed them for comparison between analog and digital IMC as follows.

The implementation of IMC is divided into digital and analog techniques. In the case of digital IMC, MAC operations are predominantly carried out using logic gates. However, this approach not only consumes a large area due to wide adder trees but also requires more clock cycles since the operations are executed step by step. On the other hand, analog IMC achieves efficient MAC operations by leveraging Ohm's law and Kirchhoff's laws [5]. This results in reduced area consumption and allows for faster computation [6].

[5] P. Yao, DOI: 10.1038/s41586-020-1942-4;

[6] I. Muñoz-Martín, doi: 10.1109/TED.2021.3118996.

  • Could the Authors provide an estimation of throughput and GOPS/W with larger size of the array (i.e., with a mathematical model and/or numerical simulations?

Response: Thanks for your comments. When calculating the throughput of fully parallel in-memory computing, it is typically computed using the formula (array size / one cycle time). However, since the conventional MAC operation involves separate multiplication and accumulation steps, the proposed in-memory computing generates a one-step VMAC. Therefore, the calculation should be multiplied by 2, resulting in (2 × array size / one cycle time). The dominant power consumption in the proposed architecture is attributed to the R-ladder. In the overall proposed structure, it consumes 3.04mW up to the output of the ADC, and the power consumption of the 8 R-ladders, each corresponding to an ADC, accounts for 2mW, constituting approximately 66% of the total power.

Therefore, in a 128×128 array size, 32 ADCs are utilized, and the R-ladder consumes 8mW. Applying the proportion of R-ladder power consumption to the overall power consumption in the existing array size, the average power consumption for a 128×128 array size is calculated to be 12.12mW, achieving 135.2 TOPS/W.

The above content has been summarized and added to Section 4.4 as follows.

In the proposed architecture, the dominant power consumption is due to the R-ladder. The dynamic power consumption of eight R-ladders, each corresponding to an ADC, is 2mW, accounting for approximately 66% of the total power. For example, if the array size of the proposed architecture is expanded to 128 x 128, 32 ADCs are used, and 32 R-ladders consume 8 mW of dynamic power. When R-ladder is accounted for, the average power consumption is calculated to be 12.12 mW, and the throughput and TOPS/W are 1638.4 GOPS and 135.2 TOPS/W, respectively.

Round 2

Reviewer 1 Report

Comments and Suggestions for Authors

The authors have solved all the issues in the last version of the manuscript. The manuscript can be accepted now.